# Hypertension and Low Body Weight Are Associated with Depressive Symptoms Only in Females: Findings from the Shika Study

**DOI:** 10.3390/bs12110413

**Published:** 2022-10-27

**Authors:** Toru Yanagisawa, Fumihiko Suzuki, Hiromasa Tsujiguchi, Akinori Hara, Sakae Miyagi, Takayuki Kannon, Keita Suzuki, Yukari Shimizu, Thao Thi Thu Nguyen, Fumika Oku, Kuniko Sato, Masaharu Nakamura, Koichiro Hayashi, Aki Shibata, Tadashi Konoshita, Yasuhiro Kambayashi, Hirohito Tsuboi, Atsushi Tajima, Hiroyuki Nakamura

**Affiliations:** 1Department of Public Health, Graduate School of Advanced Preventive Medical Sciences, Kanazawa University, 13-1 Takaramachi, Kanazawa 920-8640, Japan; 2Department of Hygiene and Public Health, Faculty of Medicine, Institute of Medical, Pharmaceutical and Health Sciences, Kanazawa University, Kanazawa 920-8640, Japan; 3Community Medicine Support Dentistry, Ohu University Hospital, Koriyama 963-8611, Japan; 4Advanced Preventive Medical Sciences Research Center, Kanazawa University, 1-13 Takaramachi, Kanazawa 920-8640, Japan; 5Innovative Clinical Research Center, Kanazawa University, 13-1 Takaramachi, Kanazawa 920-8641, Japan; 6Department of Bioinformatics and Genomics, Graduate School of Advanced Preventive Medical Sciences, Kanazawa University, 13-1 Takaramachi, Kanazawa 920-8640, Japan; 7Faculty of Health Sciences, Department of Nursing, Komatsu University, 14-1 Mukaimotorimachi, Komatsu 923-0961, Japan; 8Faculty of Public Health, Hai Phong University of Medicine and Pharmacy, Ngo Quyen, Hai Phong 180000, Vietnam; 9Department of Clinical Cognitive Neuroscience, Graduate School of Medical Science, Kanazawa University, Kakuma-machi, Kanazawa 920-1192, Japan; 10Third Department of Internal Medicine, University of Fukui Faculty of Medical Sciences, 23-3 Matsuoka Shimoaizuki, Eiheiji-cho, Yoshida-gun, Fukui 910-1193, Japan; 11Department of Public Health, Faculty of Veterinary Medicine, Okayama University of Science, 1-3 Ikoinooka, Imabari 794-8555, Japan; 12Institute of Medical, Pharmaceutical and Health Sciences, Kanazawa University, 1 Kakuma-Machi, Kanazawa 920-1192, Japan

**Keywords:** blood pressure, body mass index, geriatric depression scale, cross-sectional study, multiple regression analysis

## Abstract

Although the relationship between hypertension and depression is influenced by several physiological factors, including body weight and other lifestyle factors, such as eating behavior, the specific involvement of depression in hypertension remains unclear. Therefore, this epidemiological study examined the role of body weight in the relationship between hypertension and depressive symptoms among the middle-aged and elderly living in the community of Shika town. In total, 1141 males and 1142 females with mean ages of 69.09 and 70.61 years, respectively, participated this study. Physiological factors, including blood pressure, body mass index (BMI), and lifestyle, were investigated in a medical check-up in Shika town. Depressive symptoms were evaluated using the Geriatric Depression Scale 15 (GDS-15). A two-way analysis of covariance exhibited a significant interaction between the two hypertensive groups and body size groups on GDS in females. The post hoc Bonferroni method showed that in the hypertensive groups, GDS was significantly higher in the underweight group (BMI < 18.5) than in the standard/overweight group; however, this relationship was not observed in the no-hypertensive groups. Multiple regression analysis also verified this relationship. Therefore, it is suggested that the combination of hypertension and being underweight is associated with depressive symptoms only in females.

## 1. Introduction

According to the National Center for Health Statistics in 2019 [1], 18.5% of adults had depressive symptoms that were either mild, moderate, or severe during the past two weeks. Depression has been found to be a significant comorbidity of chronic diseases, such as chronic obstructive pulmonary disease [2], asthma [3], and rheumatoid arthritis [4]. Another of these comorbidities is hypertension [5,6,7]. In a meta-analysis of 41 studies by Li et al. [8], the prevalence of depression in patients with hypertension was 26.8%. Although the relationship between hypertension and depression is considered to be influenced by low physical activity [9], stress-induced improves in sympathetic nerve activity, and other lifestyle changes, including the consumption of large amounts of alcohol [5], the possible involvement of other modulators, such as low body weight, in the relationship between hypertension and depression remains unclear.

An important factor related to depressive symptoms is body weight [10,11,12]. A longitudinal study using cross-lagged panel models by Kim et al. [10] showed that low body weight was associated with depressive symptom scores. Alternatively, meta-analyses by Luppino et al. [11] reported a reciprocal link between depression and being overweight. Additionally, a narrative review by Al-Khatib et al. [13] reported a relationship between depression and metabolic syndrome. Furthermore, a meta-analysis by Jung et al. [12] reported that being underweight and overweight were associated with depression despite differences in ethnicity and sex. However, in a meta-analysis of cohort studies, Jung et al. [12] reported that low body weight at baseline was associated with subsequent depression development, whereas the same analysis found no significant difference in overweight individuals. Kim et al. [10] also reported that depression caused weight loss rather than gain in middle-aged and elderly Asian populations. Being underweight is partly caused by poor nutrition; therefore, low levels of leptin, a mood change-related mediator, have been implicated in the depression development [14]. Therefore, it is speculated that low body weight is more likely associated with depression than being overweight in middle-aged and elderly Japanese individuals.

Although the relationship between hypertension and depression [8,9] or between body size and depression [10,11,12] has been examined, whether the combination of hypertension and body size further increases depressive symptoms has not been studied in detail. Since hypertension and obesity correlate positively [15], a similar mechanism may affect depression. Alternatively, hypertension and low body weight affect depression through various mechanisms. It has been reported that undernutrition is associated with depression in underweight [16], whereas lifestyle factors such as stress and poor sleep are related to non-obese hypertension [17]. Therefore, it is hypothesized that hypertension combined with low body weight intensifies depressive symptoms through various mechanisms. Therefore, this cross-sectional study investigated the role of low body weight in the relationship between hypertension and depressive symptoms among middle-aged and elderly individuals living in the community Shika town.

## 2. Materials and Methods

### 2.1. Study Design and Participants

This cross-sectional study, conducted among the residents of Shika town, was termed the Shika study [18,19,20]. Participants were recruited between October 2013 and December 2016. The target population was the residents of Shika town, Ishikawa Prefecture, Japan (population, 21,301, males, 10,111, females 11,190, number of individuals aged 65 years and older on 1 December, 2016, 8498 (aging rate 39.9%) [21]. Shika town has a slightly older population than the national average, with no substantial variation in population composition or sex ratio, making it a representative epidemiological population of rural Japan [22]. Five thousand one hundred and thirteen residents aged 40 years and older live in the four model districts (Horimatsu, Higashimasuho, Tsuchida, and Togi districts). Written informed consent was collected from all 5013 participants. Of these, 3139 participants were aged 55 years. We excluded 257 participants because they were receiving treatment for depression, in addition to 299 who had fewer than ten responses to the Geriatric Depression Scale 15 (GDS-15). Therefore, 2583 participants (1141 males and 1442 females) were included in this analysis (Figure 1).

Abbreviations: GDS, Geriatric Depression Scale.

### 2.2. Instrumentation

#### 2.2.1. Blood Pressure Assessment

In a medical check-up in Shika town, well-trained nurses and clinical technologists measured blood pressure for all participants. Blood pressure was measured twice consecutively using the right upper arm with an appropriately sized cuff attached to UM-15P (Parama-tech Co., Ltd., Fukuoka, Japan) and HEM-907 (OMRON Co., Ltd., Kyoto, Japan), an automated digital sphygmomanometer based on the oscillometric method. The mean values of these measurements were used in the analyses. Hypertension was defined as systolic blood pressure ≥ 140 mmHg and/or diastolic blood pressure ≥ 90 mmHg. Participants receiving antihypertensive treatment were included in this study.

#### 2.2.2. Body Size Assessment

The body sizes of participants were classified as follows based on the body mass index (BMI) collected from medical check-up data in Shika town: underweight, BMI < 18.5; standard weight, BMI > 18.5 and <25; overweight, BMI > 25 [23].

### 2.3. Assessment of Depressive Symptoms

Depressive symptoms were assessed using the Japanese short version of GDS-15, comprising 15 questions developed for self-report surveys [24]. Each item is rated in a yes/no format. Among them, ten items (2, 3, 4, 6, 8. 9, 10, 12, 14, and 15) showed depression when answered “yes” (positive), while the remaining five items (1, 5, 7, 11, and 13) showed depression when answered “no” (negative). The total possible score ranged from 0–15. Higher scores show more severe depressive symptoms. The validity and reliability of the GDS-15 Japanese version for depression evaluated against the Diagnostic and Statistical Manual of Mental Disorders, Fourth Edition, Text Revision (DSM-IV-TR) criteria have already been verified [25]. When the cutoff point was set at 6/7, it has been confirmed that the sensitivity was 0.98, specificity was 0.86, and Cronbach’s alpha reliability coefficient was 0.83 [25].

### 2.4. Questionnaire on Demographics

Participants completed self-report questionnaires on age (years), occupation/volunteer status (yes/no), marital status (yes/no), living alone (yes/no), exercise/hobbies (yes/no), drinking status (habitual drinker/non-drinker or rarely drinker), smoking status (current smoker/non-smoker or past smoker), and diabetes diagnosis at a local hospital.

### 2.5. Statistical Analysis

Participants were categorized into the hypertensive and no-hypertensive groups. Participants were also divided into the following two groups: underweight and standard weight/overweight groups. IBM SPSS Statistics version 25 for Windows (IBM, Armonk, NY, USA) was used for statistical analyses. The Student’s *t*-test was conducted to investigate the relationships between continuous variables, while the chi-square test was used to examine the relationships between categorical variables. A two-way analysis of covariance (ANCOVA) adjusted for age, occupation/volunteer, marital status, living alone, exercise/hobbies, drinking status, smoking status, diabetes, and hypertension treatment was conducted to investigate the main effects and interactions between the two hypertensive groups and two body size groups on depressive symptoms. A multiple regression analysis was used to investigate the relationship between hypertension and low body weight on depressive symptoms to verify the results of the two-way ANCOVA. Participants were stratified into hypertensive and no-hypertensive groups and evaluated according to sex. The forced input method was used for variable selection. Pairwise deletion was used to handle missing values in the data. The significance level was set at 5%.

### 2.6. Sample Size and Statistical Power

Free G-power software was used to calculate the sample size and statistical power. For the F-tests of ANCOVA, the effect size, alpha error probability, power, number of groups, and number of covariates were set to 0.25, 0.05, 0.95, 4, and 9, respectively. The total sample size and actual power were 210 and 0.950, respectively. For the F-tests for multiple linear regression, the effect size, alpha error probability, power, and many of the predictors were set to 0.15, 0.05, 0.95, and 9, respectively. The total sample size and actual power were 166 and 0.950, respectively. Therefore, the sample size of this study was verified to be sufficient.

## 3. Results

### 3.1. Participant Characteristics

Table 1 shows participant features. Among 2583 participants, 1141 were males, and 1442 were females. Males were significantly younger than females (mean ± standard deviation [SD]; 69.09 ± 8.47 vs. 70.61 ± 9.74 years, *p* < 0.001). The proportions of occupation/volunteer (*p* < 0.001), with and without a spouse (*p* < 0.001), living alone (*p* < 0.001), exercise/hobby (*p* = 0.007), and being underweight (*p* < 0.001) were significantly higher among females than males. Alternatively, the proportions of current drinkers (*p* < 0.001), current smokers (*p* < 0.001), diabetes (*p* < 0.001), and hypertension (*p* = 0.001) were significantly higher among males than among females. No significant sex differences were observed in GDS.

### 3.2. Characteristics of GDS Based on Hypertension by Sex

Table 2 shows participant features subclassified by hypertension and GDS based on sex. There were 583 male and 905 female participants in the no-hypertensive groups and 558 and 537, respectively, in the hypertensive groups. GDS insignificantly differ between males and females with hypertension.

### 3.3. Characteristics of Body Weight and GDS by Sex

Table 3 shows participant features subclassified by body weight and GDS based on sex (Table 3). There were 40 male and 123 female participants in the underweight groups, and 1101 and 1319, respectively, in the standard/overweight groups. GDS was significantly higher in the underweight group than in the standard/overweight group for males (*p* = 0.013) and females (*p* < 0.001), respectively.

### 3.4. Interaction between Hypertension and Body Size on GDS

Table 4 shows the results of the two-way ANCOVA on GDS subclassified by hypertension and body size based on sex. Covariates were adjusted for age, occupation/volunteer, with and without a spouse, living alone, exercise/hobby, current drinker, current smoker, diabetes, and hypertension. There was no main effect or interaction for GDS in males between hypertension and body size. Alternatively, there was a significant main effect of GDS in females between the hypertensive groups (*p* = 0.014) and body size groups (*p* < 0.001) (Table 5). There was also a significant interaction between the hypertensive groups and body size groups on GDS in females (*p* = 0.049). The post hoc analysis using the Bonferroni technique indicated that in the hypertensive groups, GDS was significantly higher in the underweight group than in the standard/overweight group (*p* = 0.002); however, this relationship was not observed in the no-hypertensive groups. The Bonferroni technique also demonstrated that GDS was significantly higher in underweight participants with hypertension than in the other three groups. Therefore, the combination of hypertension and low body weight is synergistically related to depressive symptoms in the females but not in males.

### 3.5. Multiple Regression Analysis of Low Body Weight and Confounding Factors on GDS Stratified by Hypertension in Females

Table 6 demonstrates the results of a multiple regression analysis of low body weight and confounding factors on GDS stratified by hypertension in females. In the no-hypertensive group, age (partial regression coefficient (95% confidence interval); *p*-value) (0.029 (0.001, 0.057); *p* = 0.041) and exercise/hobby (−1.093 (−1.565, −0.621); *p* < 0.001) significantly contributed to GDS as independent variables. In contrast, in the hypertensive group, age (0.078 (0.038, 0.118); *p* < 0.001), exercise/hobby (−0.182 (−1.846, −0.549); *p* < 0.001), current smoker (3.251 (0.654, 5.849); *p* = 0.041), and a low body weight (2.017 (0.761, 3.273); *p* = 0.002) were significant independent variables. Therefore, the multivariate analysis also demonstrated that females with low body weight were associated with depressive symptoms in hypertension but not in normotension.

## 4. Discussion

Females in the no-hypertensive group, GDS did not differ significantly between the body size groups. In contrast, in females in the hypertensive group, it was significantly higher in the underweight group than in the standard/overweight group. In a cross-sectional study of middle-aged and elderly primary care populations at risk of cardiovascular disease, Rantannen et al. [5] found the factors predisposing hypertensive patients to depression as female sex, harmful alcohol use, being overweight, smoking, and low levels of leisure-time physical activity. Additionally, a cross-sectional study of countryside-dwelling older Chinese individuals by Ma [6] showed that depression in conjunction with hypertension was associated with factors including female sex, being unmarried, and living alone. A review by Scalco et al. [9] that examined the relationship between hypertension and depression exhibited significantly higher noradrenaline levels in patients with essential hypertension than in normotensives [26] and higher noradrenaline levels during orthostasis and exercise in depressed patients than in normotensives [27]. Alternatively, a review by Mendoza et al. [15] discussed the significance of adiposity in obesity hypertension centers on humoral mechanisms through stimulation of the renal-angiotensin system, leptin activity, sympathetic overdrive, and pro-inflammatory processes. Although different factors may be involved in the influence of hypertension on depression, such as physical inactivity, unhealthy lifestyles, including excessive alcohol consumption, and increased sympathetic nerve activity owing to stress, it is necessary to separate these factors in obese and non-obese hypertension. Our findings that showed a significant correlation with GDS in underweight hypertensive females suggest that further investigations are required to determine which factors may synergistically exacerbate depressive symptoms.

Being underweight or overweight has both been associated with depression [12]. Additionally, a cross-sectional study by Hadi et al. reported that central obesity mediates hypertension and depression [28]. A cross-sectional study by Yu et al. [29] speculated that the negative body image of underweight individuals was related to low self-esteem and depressive symptoms. Alternatively, a review by Remes et al. [30] reported that the relationship between body size and depression differed with race, country, sex, age, and time. A longitudinal study by Kim et al. [10] also showed that depression may induce weight loss rather than gain in middle-aged and elderly Asian populations. Although previous studies on body size and depression have implicated underweight and overweight, a difference seems to be because of racial influences. Therefore, it was considered meaningful to conduct epidemiological studies on the Japanese of Asian descent. A large-scale cross-sectional study by Liao et al. [31] reported that the relationship between low body weight and depressive symptoms was common in females and adolescents. A longitudinal study of the relationship between BMI and depression in the elderly by Kim et al. [10] reported that depressive symptom scores were higher in the underweight group than in the normal weight group over time. As a possible mechanism underlying the relationship between low body weight and depression, Lang et al. [32] showed that leptin was a mediator related to energy homeostasis and mood changes in the gut-brain circuit. A clinical study [14] showed the antidepressant effects of leptin. Additionally, since neuropeptide Y, which comprises 36 amino acids, is involved in the relationship between weight gain and the attenuation of depressive symptoms [33], a decrease in its neurotransmitters may be related to depression. Therefore, it is speculated that low leptin and amino acid levels owing to poor nutrition may be contributing factors to depression development. These results showed that depressive symptoms were associated only with hypertension in the underweight group. Therefore, in addition to the relationship between hypertension and depression, it is speculated that poor nutrition, a contributing factor to low body weight, causes low levels of leptin and amino acids, resulting in more severe depressive symptoms. Alternatively, the reason for the lack of this relationship in the standard/overweight group may be because of the suppressive effects of sufficient leptin and amino acids on depressive symptoms.

The relationship between body size and depression is still unclear because of the involvement of several factors. In our preliminary analysis, no significant differences were found in GDS between standard weight and overweight groups. It is speculated that the lack of a relationship between being overweight and depression may be due to leptin absence [14] or neuropeptide Y [33], which are associated with depressive symptoms. The findings of our previous Shika study showed that a lower vitamin [34] or n-3 polyunsaturated fatty acid [35] intake was associated with depressive symptoms in overweight women. However, since these nutrients were assessed relative to the percentage of daily caloric intake, it is presumed that absolute amounts in overweight women with a high daily intake are not deficient. Therefore, further research on the relationship between depression and body size needs to include a nutritional assessment.

These results showed that GDS was significantly higher in females only in the hypertensive and underweight groups. A reason for this sex difference is that, in menopausal women, a reduction in sex hormones is associated with hypertension [36,37,38] and depression [39,40]. An investigation of healthy Polish women in various stages of menopause by Cybulska et al. [41]. showed that specific genotypes are related to depressive symptoms. Another possible reason for low body weight is sarcopenia, a loss of muscle mass. Cross-sectional studies by Lee et al. [42] and Sun et al. [43] showed a relationship between sarcopenia and depression in women only. Therefore, we speculate that the mechanism why only females with hypertension and low body weight have worsening depressive symptoms may be because of the reduced leptin associated with inadequate nutrition, the increased blood pressure lifestyle with non-obesity, and the reduction in estrogen associated with menopause, synergistically contributing to depression.

### Limitations and Future Directions

Several limitations need to be addressed. Since this was a cross-sectional study, further longitudinal studies are required to elucidate the causal relationship between hypertension, low body weight, and depressive symptoms. Furthermore, nutrient intake and estrogen levels were not evaluated. Another limitation was that depressive symptoms other than GDS were not evaluated. Moreover, depression was not diagnosed. Lastly, many participants are needed in the overweight group for future analyses.

Future research should elucidate the specific combination of factors among low nutrition, lifestyle habits that increase blood pressure, and low estrogen levels that synergistically exacerbate depressive symptoms. Furthermore, it is necessary to evaluate longitudinal studies to determine whether any combination of nutritional guidance, lifestyle guidance for hypertension, and estrogen replacement therapy can effectively alleviate depressive symptoms.

## 5. Conclusions

In females with hypertension, GDS was significantly higher in the underweight group than in the standard/overweight group, whereas no relationship was found in the no-hypertensive groups. As a possible mechanism, in underweight females with hypertension, reduced leptin, increased blood pressure lifestyle, and a reduction in estrogen levels are synergistically associated with depressive symptoms. Further longitudinal analyses are required to validate this relationship.

## Figures and Tables

**Figure 1 behavsci-12-00413-f001:**
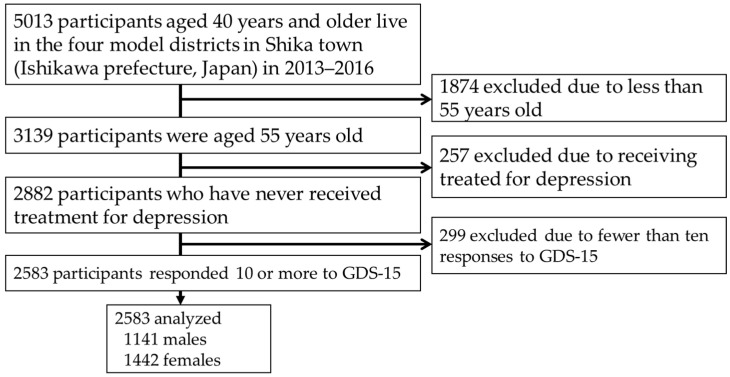
Participant recruitment chart.

**Table 1 behavsci-12-00413-t001:** Participant characteristics.

	Male (*n* = 1141)	Female (*n* = 1442)	*p*-Value ^a^
Age, years	69.09 ± 8.47	70.61 ± 9.74	**<0.001**
GDS	4.00 ± 3.39	3.85 ± 3.16	0.262
Occupation/volunteer, *n (%)*	393 (34.44)	402 (27.88)	**<0.001**
Without spouse, *n* (%)	156 (13.67)	454 (31.48)	**<0.001**
Living alone, *n* (%)	91 (7.98)	192 (13.31)	**<0.001**
Exercise/hobby, *n* (%)	449 (39.35)	491 (34.05)	**0.007**
Current drinker, *n* (%)	911 (79.84)	754 (52.29)	**<0.001**
Current smoker, *n* (%)	325 (28.48)	53 (3.68)	**<0.001**
Diabetes, *n* (%)	169 (14.81)	131 (9.08)	**<0.001**
Hypertension, *n* (%)	558 (48.90)	537 (47.06)	**0.001**

^a^*p*-values were calculated using the Student’s *t*-test for continuous variables and the chi-square test for categorical variables (*p*-values less than 0.05 are highlighted in bold). Data are shown as mean ± standard deviation or n numbers and percentages. Abbreviations: SD, standard deviation; GDS, Geriatric Depression Scale.

**Table 2 behavsci-12-00413-t002:** Characteristics of hypertension and GDS according to sex.

	No-Hypertensive Group	Hypertensive Group	*p*-Value ^a^
	Mean (95% CI)	Mean (95% CI)	
Male (*n* = 1141)	4.07 (3.82, 4.32)	3.86 (3.54, 4.18)	0.302
Female (*n* = 1442)	3.83 (3.63, 4.03)	3.89 (3.61, 4.17)	0.710

^a^*p*-values were calculated using the Student’s *t*-test. Abbreviations: CI, confidence interval; GDS, Geriatric Depression Scale.

**Table 3 behavsci-12-00413-t003:** Characteristics of body weight and GDS according to sex.

	Underweight Group	Standard/Overweight Group	*p*-Value ^a^
	Mean (95% CI)	Mean (95% CI)	
Male (*n* = 1141)	5.30 (4.08, 6.52)	3.95 (3.75, 4.15)	**0.013**
Female (*n* = 1442)	5.14 (4.53, 5.75)	3.73 (3.56, 3.90)	**<0.001**

^a^*p*-values were calculated using the Student’s *t*-test (*p*-values less than 0.05 are highlighted in bold). Abbreviations: CI, confidence interval; GDS, Geriatric Depression Scale.

**Table 4 behavsci-12-00413-t004:** Interactions between body size and hypertension on GDS in males.

	No-Hypertensive Group	Hypertensive Group	*p*-Value ^a^		
	Mean (95% CI) (*n*)	Mean (95% CI) (*n*)	*P*1	*P*2	*P*3
Underweight group	5.00 (3.26, 6.74) (*n* = 25)	5.36 (3.15, 7.58) (*n* = 15)	0.343	0.282	0.350
Standard/overweight group	3.99 (3.72, 4.26) (*n* = 558)	3.77 (3.43, 4.10) (*n* = 543)			

^a^ Analysis of covariance. Adjusted for age, occupation/volunteer, without a spouse, living alone, exercise/hobby, current smoker, diabetes, and hypertension. *P*1: body size groups, *P*2: hypertensive groups, *P*3: body size × hypertensive groups. Abbreviations: CI, confidence interval; GDS, Geriatric Depression Scale.

**Table 5 behavsci-12-00413-t005:** Interactions between body size and hypertension on GDS in females.

	No-Hypertension	Hypertension	*p*-Value ^a^		
	Mean (95% CI) (*n*)	Mean (95% CI) (*n*)	*P*1	*P*2	*P*3
Underweight	4.47 (3.70, 4.76) (*n* = 96)	6.09 (4.76, 7.42) (*n* = 27)	**<0.001**	**0.014**	**0.049**
Standard/overweight	3.60 (3.37, 3.83) (*n* = 809)	3.68 (3.36, 3.99) (*n* = 510)			

^a^ Analysis of covariance. Adjusted for age, occupation/volunteer, without a spouse, living alone, exercise/hobby, current smoker, diabetes, and hypertension. *P*1: body size groups, *P*2: hypertensive groups, *P*3: body size × hypertensive groups. Abbreviations: CI, confidence interval; GDS, Geriatric Depression Scale.

**Table 6 behavsci-12-00413-t006:** Relationship between underweight and GDS stratified by hypertension in females.

		Standardized Coefficient	95% CI		*p*-Value ^a^
		β	Lower	Upper	
No-hypertensive group	Age	0.090	0.001	0.057	**0.041**
(*n* = 1007)	Exercise/hobby (none)	−0.169	−1.565	−0.621	**<0.001**
	Current smoker	−0.016	−1.333	0.849	0.663
	Underweight	0.063	−0.088	1.337	0.086
Hypertensive group	Age	0.244	0.038	0.118	**<0.001**
(*n* = 435)	Exercise/hobby (none)	−0.182	−1.846	−0.549	**<0.001**
	Current smoker	0.123	0.654	5.849	**0.014**
	Underweight	0.156	0.761	3.273	**0.002**

^a^ Multiple regression analysis (*p*-values less than 0.05 are highlighted in bold). Independent variables were adjusted for occupation/volunteer, living alone without a spouse, current drinker, and diabetes. Abbreviations: CI, confidence interval; GDS, Geriatric Depression Scale.

## Data Availability

Data in the present study are available upon request from the corresponding author. Data are not publicly available due to privacy and ethical policies.

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
