# Peer review of "Hypertension and Low Body Weight Are Associated with Depressive Symptoms Only in Females: Findings from the Shika Study"

_behavsci, 2022, doi:10.3390/bs12110413_

Round 1

Reviewer 1 Report

This is well written and an overall good study technically. However, the significance of this study has not been established. Why do we need to study this relationship in this population? Why do we need to study this relationship at all?? How will the findings benefit human populations? Why is this particular study needed and what gaps does it fill?

Content

Introduction:

 Be exact with your statements. In the introduction, what involvement are you referring to in this statement “the 64 exact involvement of depression in hypertension remains unclear.” Does the relationship need to be explained, do we need to explore low body weight as a mediators, or moderators of this relationship, do we need to know if low body weight is a predictor or intervening variable? Be specific instead of just saying what is the involvement. What are the specific gaps in the literature about how this relationship?

Significance of the study. You do not establish the significance of examining populations in Shika town. It this a gap in the literature? What population are you studying and why? What age group? What gender? Race/ethnicity? Essentially why is it important to study this relationship in this population of people? Males, females, etc. You need a literature review that outlines the gaps for your population and how this study fills that gap. Why is this population targeted?

You did not define who shika town is??

Methods: You Need further description of the sample and sample selection. What is the inclusion and exclusion criteria?

Methods: Was this a secondary analysis of existing data? If so say that at part of the study methods.

You state. Hypertension was defined as blood 107 pressure ≥140/90 mmHg. What if one number was high and the other number normal? Then how was Hypertension defined?

Did you calculate any coefficients of variations for any of the instruments?

The measurements require more descriptions.

Discuss the psychometric properties of the GDS-15 Japanese version and what populations has the psychometric properties been assessed in? What is the reliability in this sample?

More description of the scales are needed. What kind of scale is this? Likert-type? Ordinal? Are there any subscales? Do you use a total score or subscale scores separately?

Instead of saying  self-administered say “self-report” instruments.

How were these variables measured? age, occupation/volun- 124 teer status, marital status, living alone, exercise/hobbies, drinking status, smoking status, 125 and diabetes diagnosis.

For example was age measure continuously or categorically, what are the occupational categories, the marital categories, smoking status (no smoking, heavy smoker, light smoker etc.)

Would be nice to have clinical implications/recommendations

Mechanics

Be parsimonious. You don’t need unnecessary phrases. Be direct and to the point for better precision and clarity for example, in the introduction, you don’t need this beginning statement, “Regarding the relationship between hypertension and depression” Just start with only a few studies.

Under the discussion same problem. You don’t need to say “The main result of the present study was that in… Just start with “females in the no-hypertension”

You need a subheading fo “Procedures for Data Collection” and under this a description about how the rights of human subjects were protected. Was an IRB involved? Was informed consent obtained?

Organization:

You need to insert headings and subheadings.

You need a heading that says “Instrumentation” before describing the instruments.

Insert a heading that says “limitations” before the limitations discussion

You need a subheading that says “design”

Author Response

Reviewer 1

Comments and Suggestions for Authors

This is well written and an overall good study technically. However, the significance of this study has not been established. Why do we need to study this relationship in this population? Why do we need to study this relationship at all?? How will the findings benefit human populations? Why is this particular study needed and what gaps does it fill?

Comment 1

Introduction:

Be exact with your statements. In the introduction, what involvement are you referring to in this statement “the 64 exact involvement of depression in hypertension remains unclear.” Does the relationship need to be explained, do we need to explore low body weight as a mediators, or moderators of this relationship, do we need to know if low body weight is a predictor or intervening variable? Be specific instead of just saying what is the involvement. What are the specific gaps in the literature about how this relationship?

Response 1

We have amended the introduction section of the revised manuscript as follow:

“Although the relationship between hypertension and depression is considered to be influenced by low physical activity [9], stress-induced increases in sympathetic nerve activity, and other lifestyle changes, such as the consumption of large amounts of alcohol [5], the possible involvement of other modulators, such as low body weight, in the relationship between hypertension and depression remains unclear.”

(L67-69)

Comment 2

Significance of the study. You do not establish the significance of examining populations in Shika town. It this a gap in the literature? What population are you studying and why? What age group? What gender? Race/ethnicity? Essentially why is it important to study this relationship in this population of people? Males, females, etc. You need a literature review that outlines the gaps for your population and how this study fills that gap. Why is this population targeted?

You did not define who shika town is??

Response 2

We have amended the participants section of the revised manuscript as follow:

“The target population was the residents living in Shika town, Ishikawa prefecture, Japan (population, 21301, males, 10111, females 11190, number of individuals aged 65 years and older on 1 December 2016, 8498 (aging rate 39.9%)) [18]. Shika town has a slightly older population than the national average, with no substantial differences in population composition or sex ratio, making it a representative epidemiological population for rural Japan. A total of 5013 residents aged 40 years and older live in the of 4 model districts (Horimatsu, Higashimasuho, Tsuchida, and Togi districts).” (L100-106)

Because of the high response rate and low bias of the Shika town residents to the questionnaire survey, we have published a number of articles including the Shika study in the subtitle.

https://pubmed.ncbi.nlm.nih.gov/?term=Shika+study+Kanazawa

Comment 3

Methods: You Need further description of the sample and sample selection. What is the inclusion and exclusion criteria?

Response 3

We have added Figure 1 (Participant recruitment chart) in the participants section of the revised manuscript (below line 158)

Comment 4

Methods: Was this a secondary analysis of existing data? If so say that at part of the study methods.

Response 4

No secondary data were used in this study.

Comment 5

You state. Hypertension was defined as blood pressure ≥140/90 mmHg. What if one number was high and the other number normal? Then how was Hypertension defined?

Response 5

We have amended the Blood pressure assessment section of the revised manuscript as follows:

“Hypertension was defined as systolic blood pressure ≥140 mmHg and/or diastolic blood pressure ≥90 mmHg.” (L170-171)

Comment 6

Did you calculate any coefficients of variations for any of the instruments?

Response 6

We have confirmed that the coefficient of variation for GDS is 0.83 and that for BMI is 0.14.

Comment 7

The measurements require more descriptions.

Discuss the psychometric properties of the GDS-15 Japanese version and what populations has the psychometric properties been assessed in? What is the reliability in this sample?

Response 7

Our analysis focused on the general Japanese community residents and confirmed that GDS-15 scores are similar when compared to another study of similarly Japanese subjects.

https://repository.kulib.kyoto-u.ac.jp/dspace/bitstream/2433/182937/1/j.psychres.2013.12.015.pdf

Reliability has been demonstrated in other studies; therefore, the assessment of depressive symptoms section of the revised manuscript was amended as follows:

“The validity and reliability of the GDS-15 Japanese version for depression assessed against the Diagnostic and Statistical Manual of Mental Disorders, Fourth Edition, Text Revision (DSM-â…£-TR) criteria have already been confirmed [23].” (L185-188)

Comment 8

More description of the scales are needed. What kind of scale is this? Likert-type? Ordinal? Are there any subscales? Do you use a total score or subscale scores separately?

Response 8

We have amended the assessment of depressive symptoms section of the revised manuscript as follows:

“Each item is rated in a yes/no format. Among them, 10 items (2, 3, 4, 6, 8. 9, 10, 12, 14, and 15) indicate the presence of depression when answered “yes” (positive), while the remaining 5 items (1, 5, 7, 11, and 13) indicated depression when answered “no” (negative). The potential total score ranged from 0 to 15.” (L181-185)

Comment 9

Instead of saying self-administered say “self-report” instruments.

Response 9

We have amended the Materials and Methods section of the revised manuscript as follows:

“Depressive symptoms were evaluated using the Japanese short version of GDS-15, consisting of 15 questions developed for self-report surveys [22].” (L180-181)

Comment 10

How were these variables measured? age, occupation/volun- 124 teer status, marital status, living alone, exercise/hobbies, drinking status, smoking status, 125 and diabetes diagnosis.

For example was age measure continuously or categorically, what are the occupational categories, the marital categories, smoking status (no smoking, heavy smoker, light smoker etc.)

Would be nice to have clinical implications/recommendations

Response 10

We have amended the Questionnaire on demographics section of the revised manuscript as follows:

“Participants completed self-report questionnaires on age (years), occupation/volunteer status (yes/no), marital status (yes/no), living alone (yes/no), exercise/hobbies(yes/no), drinking status (habitual drinker/non-drinker or rarely drinker), smoking status (current smoker/non-smoker or past smoker), and diabetes diagnosis at a local hospital.” (L200-203)

Comment 11

Mechanics

Be parsimonious. You don’t need unnecessary phrases. Be direct and to the point for better precision and clarity for example, in the introduction, you don’t need this beginning statement, “Regarding the relationship between hypertension and depression” Just start with only a few studies.

Response 11

We have amended the introduction section of the revised manuscript as follows:

“Only a few studies have investigated whether a low body weight synergistically exacerbates depressive symptoms [15].” (L89-90)

Comment 12

Under the discussion same problem. You don’t need to say “The main result of the present study was that in… Just start with “females in the no-hypertension”

Response 12

We have amended the discussion section of the revised manuscript as follows:

“Females in the no-hypertension group, GDS did not significantly differ between the body size groups, whereas in females in the hypertension group, it was significantly higher in the underweight group than in the standard/overweight group.” (L347-349)

Comment 13

You need a subheading fo “Procedures for Data Collection” and under this a description about how the rights of human subjects were protected. Was an IRB involved? Was informed consent obtained?

Response 13

We presented the information including the "Institutional Review Board Statement" or the "Informed Consent Statement" at the bottom of the conclusions section from the original file, using the template file format as follows:

“Institutional Review Board Statement: This study was conducted according to the guidelines laid down in the Declaration of Helsinki and all procedures involving research study participants were approved by the Ethics Committee of Kanazawa University (No. 1491).

Informed Consent Statement: Written informed consent was obtained from all participants prior to their participation.” (L480-484)

Comment 14

Organization:

You need to insert headings and subheadings.

You need a heading that says “Instrumentation” before describing the instruments.

Response 14

We have added the “Instrumentation” heading at L163.

Comment 15

Insert a heading that says “limitations” before the limitations discussion

Response 15

We have added the “Limitations and Future Directions” heading at L431.

Comment 16

You need a subheading that says “design”

Response 16

We have amended the section 3.1 of the revised manuscript as follows:

“3.1. Study Design and Participants

This cross-sectional study, conducted among the residents of Shika town, was termed the Shika study [16-18].” (L97-99)

Reviewer 2 Report

Dear Author,

Thank you for the opportunity to review the manuscript “ Hypertension and a low body weight are associated with depressive symptoms in females: Findings from the Shika study”. This epidemiological study investigated the role of body weight in the relationship between hypertension and depressive symptoms among middle-aged and elderly individuals living in the community of Shika town

I make the following recommendations to improve the manuscript:

1)       Page 1, line 1: Please remove “Type of the Paper”

2)       TITLE: The aim of your study was investigated the role of body weight in the relationship between hypertension and depressive symptoms among middle-aged and elderly individuals living in the community of Shika town. In title you wrote  only about females, but a total of 1141 males participated in the present study.  Please think about concise, specific and relevant title.  Maybe some question?

3)       ABSTRACT:

a.        Please indicate the main conclusions.

4)       INTRODUCTION:

·         A more elaborate introduction should be made. Explain the impact of body mass and hypertension on the depressive symptoms

·         You should describe the role of hypertension with regard to depressive symptoms, especially in postmenopausal women. Menopause is an important event in a woman's life associated with hormonal changes that play a substantial role in the functioning of her body. A decline in the level of estrogen contributes to depressive symptoms and mood disorders during this period. The severity of depressive symptoms experienced by middle-aged women depends on many factors, including sociodemographic data, genetic variables etc. Moreover postmenopausal women have hight risk of Metabolic syndrome (hypertension is one of five risk factors according to criteria by the International Diabetes Federation).

·         It seems that the authors of the manuscript should update the list of literature. Please add more recent references for example:

o    Hadi AS, Lefi A, Pikir BS, Utomo B, Lusida TTE. The association of depression and central obesity on hypertension in Indonesian provinces: a path analysis of the Indonesian baseline health research 2018 data. Blood Press. 2022 Dec;31(1):187-193. 

o    Cybulska AM, Szkup M, Schneider-Matyka D, Skonieczna-Å»ydecka K, Kaczmarczyk M, Jurczak A, Wieder-Huszla S, Karakiewicz B, Grochans E. Depressive Symptoms among Middle-Aged Women-Understanding the Cause. Brain Sci. 2020 Dec 28;11(1):26. doi: 10.3390/brainsci11010026.

o    Al-Khatib Y, Akhtar MA, Kanawati MA, Mucheke R, Mahfouz M, Al-Nufoury M. Depression and Metabolic Syndrome: A Narrative Review. Cureus. 2022 Feb 12;14(2):e22153. doi: 10.7759/cureus.22153. PMID: 35308733; PMCID: PMC8920832.

·         Please explain why this research is so important.

·         Please clearly define the aim of the work

1.        MATERIAL AND METHOD:

·         It seems necessary to clarify how the representative sample was calculated and to make a flowchart of the enrollment of respondents and study procedures.

·         Please add the reasons of inclusion and exclusion respondents

·         The instruments description should include the psychometric properties and the reference to the validation studies for the Japanese population. Please add more information about standardized survey instruments.

·         Please add sentence about ethical aproval: „The study was conducted according to the guidelines of the Declaration of Helsinki and approved by the Ethics Committee of the …”.

·         Please add the reference to „Body size assessment”

·         Table 1: please explain:  SD / % . In this column is SD or %?

5)       RESULTS

·         each of the results should be mentioned in more depth, indicating references.

·         Please improve all tables. Tables should be easy to interpret and understand.

6)       DISSCUSION:

·         The discussion of the results should be done in a more in-depth way.

·         I suggest to  prepare some kind of summary with gaps indication and future perspective

7)       REFERENCES: Please carefully check the references cited in the
manuscript.
I suggest conducting a new literature review. This article needs the newest references (written after 2020).

8)       LIMITATION: please add limitation like subsection.

·         Please discuss widely the advantages and disadvantages of this review, identify knowledge gaps in the existing literature and try to indicate the needs for future research.

9)       Since some language mistakes mainly typos are in the text, I suggest English editing before publishing of this manuscript

Author Response

Reviewer 2

Comments and Suggestions for Authors

Dear Author,

Thank you for the opportunity to review the manuscript “ Hypertension and a low body weight are associated with depressive symptoms in females: Findings from the Shika study”. This epidemiological study investigated the role of body weight in the relationship between hypertension and depressive symptoms among middle-aged and elderly individuals living in the community of Shika town

Comment 1

I make the following recommendations to improve the manuscript:

1) Page 1, line 1: Please remove “Type of the Paper”

Response 1

We have removed “Type of the Paper” in the revised manuscript at L1.

Comment 2

2) TITLE: The aim of your study was investigated the role of body weight in the relationship between hypertension and depressive symptoms among middle-aged and elderly individuals living in the community of Shika town. In title you wrote only about females, but a total of 1141 males participated in the present study. Please think about concise, specific and relevant title. Maybe some question?

Response 2

Our analysis showed that the combination of hypertension and low body weight was significantly associated with depressive symptoms synergistically only in females, but not in males.

Since we emphasize this gender difference, we ask that you allow us to keep the title unchanged.

Comment 3

3) ABSTRACT:

  1. Please indicate the main conclusions.

Response 3

We have added the following sentence at the end of the abstract as follows:

“Therefore, it is suggested that the combination of hypertension and being underweight is asso-ciated with depressive symptoms only in females.” (L50-51)

Comment 4

4) INTRODUCTION:

  A more elaborate introduction should be made. Explain the impact of body mass and hypertension on the depressive symptoms

Response 4

We have amended the introduction section of the revised manuscript as follow:

“We hypothesize that hypertension subjects with underweight have more severe de-pressive symptoms than those who are a standard weight in Asian populations, especially Japanese. Therefore, this cross-sectional study examined the role of low body weight in the relationship between hypertension and depressive symptoms among middle-aged and elderly individuals living in the community of Shika town.” (L90-95)

Comment 5

  • You should describe the role of hypertension with regard to depressive symptoms, especially in postmenopausal women. Menopause is an important event in a woman's life associated with hormonal changes that play a substantial role in the functioning of her body. A decline in the level of estrogen contributes to depressive symptoms and mood disorders during this period. The severity of depressive symptoms experienced by middle-aged women depends on many factors, including sociodemographic data, genetic variables etc. Moreover postmenopausal women have hight risk of Metabolic syndrome (hypertension is one of five risk factors according to criteria by the International Diabetes Federation).

Response 5

We have discussed the relationship between menopause and hypertension in the Discussion section (L425-430).

Our study reports sex differences in the relationship between hypertension and low body weight on depressive symptoms. Since we did not intend to study menopause, we have only briefly mentioned the relationship between menopause and hypertension in the discussion.

Comment 6

  It seems that the authors of the manuscript should update the list of literature. Please add more recent references for example:

o  Hadi AS, Lefi A, Pikir BS, Utomo B, Lusida TTE. The association of depression and central obesity on hypertension in Indonesian provinces: a path analysis of the Indonesian baseline health research 2018 data. Blood Press. 2022 Dec;31(1):187-193.

o  Cybulska AM, Szkup M, Schneider-Matyka D, Skonieczna-Å»ydecka K, Kaczmarczyk M, Jurczak A, Wieder-Huszla S, Karakiewicz B, Grochans E. Depressive Symptoms among Middle-Aged Women-Understanding the Cause. Brain Sci. 2020 Dec 28;11(1):26. doi: 10.3390/brainsci11010026.

o  Al-Khatib Y, Akhtar MA, Kanawati MA, Mucheke R, Mahfouz M, Al-Nufoury M. Depression and Metabolic Syndrome: A Narrative Review. Cureus. 2022 Feb 12;14(2):e22153. doi: 10.7759/cureus.22153. PMID: 35308733; PMCID: PMC8920832.

Response 6

We have added a cross-sectional study by Hadi et al. as reference no.26. and a narrative review by Al-Khatib et al. as reference no.13.

Comment 7

Please explain why this research is so important.

Please clearly define the aim of the work

Response 7

This is the same as response 4.

We have amended the introduction section of the revised manuscript as follow:

“We hypothesize that hypertension subjects with underweight have more severe de-pressive symptoms than those who are a standard weight in Asian populations, especially Japanese. Therefore, this cross-sectional study examined the role of low body weight in the relationship between hypertension and depressive symptoms among middle-aged and elderly individuals living in the community of Shika town.” (L90-95)

Comment 8

  1. MATERIAL AND METHOD:

It seems necessary to clarify how the representative sample was calculated and to make a flowchart of the enrollment of respondents and study procedures.

Please add the reasons of inclusion and exclusion respondents

Response 8

We have added Figure 1 (Participant recruitment chart) in the participants section of the revised manuscript (below line 158)

Comment 9

  • The instruments description should include the psychometric properties and the reference to the validation studies for the Japanese population. Please add more information about standardized survey instruments.

Response 9

We have amended the assessment of depressive symptoms section of the revised manuscript as follows:

“Each item is rated in a yes/no format. Among them, 10 items (2, 3, 4, 6, 8. 9, 10, 12, 14, and 15) indicate the presence of depression when answered “yes” (positive), while the remaining 5 items (1, 5, 7, 11, and 13) indicated depression when answered “no” (negative). The potential total score ranged from 0 to 15.” (L181-185)

Reliability has been demonstrated in other studies; therefore, the assessment of depressive symptoms section of the revised manuscript was amended as follows:

“The validity and reliability of the GDS-15 Japanese version for depression assessed against the Diagnostic and Statistical Manual of Mental Disorders, Fourth Edition, Text Revision (DSM-â…£-TR) criteria have already been confirmed [23].” (L185-188)

Comment 10

  • Please add sentence about ethical aproval: „The study was conducted according to the guidelines of the Declaration of Helsinki and approved by the Ethics Committee of the …”.

Response 10

We presented the information including the "Institutional Review Board Statement" or the "Informed Consent Statement" at the bottom of the conclusions section from the original file, using the template file format as follows:

“Institutional Review Board Statement: This study was conducted according to the guidelines laid down in the Declaration of Helsinki and all procedures involving research study participants were approved by the Ethics Committee of Kanazawa University (No. 1491).

Informed Consent Statement: Written informed consent was obtained from all participants prior to their participation.” (L480-484)

Comment 11

  • Please add the reference to „Body size assessment”

Response 11

We have added the reference no.21.

Comment 12

  • Table 1: please explain: SD / % . In this column is SD or %?

Response 12

We have modified Table 1 for clarity.

Comment 13

5) RESULTS

  • Each of the results should be mentioned in more depth, indicating references.

Response 13

We have mentioned some interpretations of the results as follows.

“Therefore, the combination of hypertension and low body weight is synergistically related to depressive symptoms only in females, but not in males.” (L306-308)

“Therefore, the multivariate analysis also showed that females with low body weight were associated with depressive symptoms in hypertension, but not in normotension.” (L337-339)

Comment 14

  • Please improve all tables. Tables should be easy to interpret and understand.

Response 14

We have modified all Tables for clarity.

Comment 15

6) DISSCUSION:

  • The discussion of the results should be done in a more in-depth way.

Response 15

We have added the following sentence to the discussion

“In addition, a cross-sectional study by Hadi et al. reported that central obesity mediates hypertension and depression [26].” (L385-386)

“Although previous studies on body size and depression have implicated both underweight and overweight, one of the differences seems to be due to racial influences. Therefore, we considered it meaningful to conduct epidemiological studies on Japanese people belonging Asian descent.” (L391-395)

Comment 16

  • I suggest to prepare some kind of summary with gaps indication and future perspective

Response 16

We have added the Limitation and Future Directions section as follows:

“Future studies need to evaluate whether nutritional interventions could improve depressive symptoms in women with hypertension and low body weight.” (L457-458)

Comment 17

7) REFERENCES: Please carefully check the references cited in the manuscript. I suggest conducting a new literature review. This article needs the newest references (written after 2020).

Response 17

We have added newest reference: no.13, no.20, no.21, and no.26.

Comment 18

8) LIMITATION: please add limitation like subsection.

Response 18

We have added the “Limitations and Future Directions” heading at L431.

Comment 19

  • Please discuss widely the advantages and disadvantages of this review, identify knowledge gaps in the existing literature and try to indicate the needs for future research.

Response 19

We have amended the discussion in the revised manuscript as we replied in responses 15 and 16.

Comment 20

9) Since some language mistakes mainly typos are in the text, I suggest English editing before publishing of this manuscript

Response 20

The English editing of the present manuscript was done when the original manuscript was written.

Round 2

Reviewer 2 Report

Dear Author,

Thank you for the opportunity to review the manuscript “ Hypertension and a low body weight are associated with depressive symptoms in females: Findings from the Shika study”. This epidemiological study investigated the role of body weight in the relationship between hypertension and depressive symptoms among middle-aged and elderly individuals living in the community of Shika town.

The authors only partially responded to the reviewer's comments. In light of the above, please refer to the reviewer's comments one more time, and please improve the manuscript.

For example, you didn't add any psychometric properties and references to the validation studies for the Japanese population (Cronbach) or you didn't improve (upgrade) the introduction, or discussion.

Author Response

Reviewer 2 Round 2

Dear Author,

Thank you for the opportunity to review the manuscript “Hypertension and a low body weight are associated with depressive symptoms in females: Findings from the Shika study”. This epidemiological study investigated the role of body weight in the relationship between hypertension and depressive symptoms among middle-aged and elderly individuals living in the community of Shika town.

The authors only partially responded to the reviewer's comments. In light of the above, please refer to the reviewer's comments one more time, and please improve the manuscript.

For example, you didn't add any psychometric properties and references to the validation studies for the Japanese population (Cronbach) or you didn't improve (upgrade) the introduction, or discussion.

Comment 1

I make the following recommendations to improve the manuscript:

1) Page 1, line 1: Please remove “Type of the Paper”

Response 1

We have removed “Type of the Paper” in the revised manuscript at L1.

Round 2 Response 1

We have already corrected it.

Comment 2

2) TITLE: The aim of your study was investigated the role of body weight in the relationship between hypertension and depressive symptoms among middle-aged and elderly individuals living in the community of Shika town. In title you wrote only about females, but a total of 1141 males participated in the present study. Please think about concise, specific and relevant title. Maybe some question?

Response 2

Our analysis showed that the combination of hypertension and low body weight was significantly associated with depressive symptoms synergistically only in females, but not in males.

Since we emphasize this gender difference, we ask that you allow us to keep the title unchanged.

Round 2 Response 2

As we responded in Response 1 of Round 1, insisting on a title that is meaningful to sex differences, we modified it as follows:

Hypertension and low body weight are associated with depressive symptoms only in females: Findings from the Shika study

Comment 3

3) ABSTRACT:

  1. Please indicate the main conclusions.

Response 3

We have added the following sentence at the end of the abstract as follows:

“Therefore, it is suggested that the combination of hypertension and being underweight is associated with depressive symptoms only in females.” (L50-51)

Round 2 Response 3

We insist that we have corrected sufficiently.

Comment 4

4) INTRODUCTION:

  A more elaborate introduction should be made. Explain the impact of body mass and hypertension on the depressive symptoms

Response 4

We have amended the introduction section of the revised manuscript as follow:

“We hypothesize that hypertension subjects with underweight have more severe depressive symptoms than those who are a standard weight in Asian populations, especially Japanese. Therefore, this cross-sectional study examined the role of low body weight in the relationship between hypertension and depressive symptoms among middle-aged and elderly individuals living in the community of Shika town.” (L90-95)

Round 2 Response 4

We have further revised the introduction section of the revised manuscript as follows:

“Although the relationship between hypertension and depression [8,9] or between body size and depression [10-12] has been examined, whether the combination of hypertension and body size further increases depressive symptoms has not been studied in detail. Since hypertension and obesity correlate positively [15], a similar mechanism may affect depression. Alternatively, hypertension and low body weight affect depression through various mechanisms. It has been reported that undernutrition is associated with depression in underweight [16], whereas lifestyle factors such as stress and poor sleep are related to non-obese hypertension [17]. Therefore, it is hypothesized that hypertension combined with low body weight intensifies depressive symptoms through various mechanisms. Therefore, this cross-sectional study investigated the role of low body weight in the relationship between hypertension and depressive symptoms among middle-aged and elderly individuals living in the community Shika town.” (L83-94)

Comment 5

  • You should describe the role of hypertension with regard to depressive symptoms, especially in postmenopausal women. Menopause is an important event in a woman's life associated with hormonal changes that play a substantial role in the functioning of her body. A decline in the level of estrogen contributes to depressive symptoms and mood disorders during this period. The severity of depressive symptoms experienced by middle-aged women depends on many factors, including sociodemographic data, genetic variables etc. Moreover postmenopausal women have hight risk of Metabolic syndrome (hypertension is one of five risk factors according to criteria by the International Diabetes Federation).

Response 5

We have discussed the relationship between menopause and hypertension in the Discussion section (L425-430).

Our study reports sex differences in the relationship between hypertension and low body weight on depressive symptoms. Since we did not intend to study menopause, we have only briefly mentioned the relationship between menopause and hypertension in the discussion.

Round2 Response 5

We have added the following sentence in the Discussion section as follows:

“An investigation of healthy Polish women in various stages of menopause by Cybulska et al. [41]. showed that specific genotypes are related to depressive symptoms.” (L336-337)

Comment 6

  It seems that the authors of the manuscript should update the list of literature. Please add more recent references for example:

o  Hadi AS, Lefi A, Pikir BS, Utomo B, Lusida TTE. The association of depression and central obesity on hypertension in Indonesian provinces: a path analysis of the Indonesian baseline health research 2018 data. Blood Press. 2022 Dec;31(1):187-193.

o  Cybulska AM, Szkup M, Schneider-Matyka D, Skonieczna-Å»ydecka K, Kaczmarczyk M, Jurczak A, Wieder-Huszla S, Karakiewicz B, Grochans E. Depressive Symptoms among Middle-Aged Women-Understanding the Cause. Brain Sci. 2020 Dec 28;11(1):26. doi: 10.3390/brainsci11010026.

o  Al-Khatib Y, Akhtar MA, Kanawati MA, Mucheke R, Mahfouz M, Al-Nufoury M. Depression and Metabolic Syndrome: A Narrative Review. Cureus. 2022 Feb 12;14(2):e22153. doi: 10.7759/cureus.22153. PMID: 35308733; PMCID: PMC8920832.

Response 6

We have added a cross-sectional study by Hadi et al. as reference no.26. and a narrative review by Al-Khatib et al. as reference no.13.

Round 2 Response 6

We further added an epidemiological survey by Cybulska et al. as new reference no. 41.

Comment 7

Please explain why this research is so important.

Please clearly define the aim of the work

Response 7

This is the same as response 4.

We have amended the introduction section of the revised manuscript as follow:

“We hypothesize that hypertension subjects with underweight have more severe de-pressive symptoms than those who are a standard weight in Asian populations, especially Japanese. Therefore, this cross-sectional study examined the role of low body weight in the relationship between hypertension and depressive symptoms among middle-aged and elderly individuals living in the community of Shika town.” (L90-95)

Round 2 Response 7

This is the same as Round2 Response 4.

We have further revised the introduction section of the revised manuscript as follows:

“Although the relationship between hypertension and depression [8,9] or between body size and depression [10-12] has been examined, whether the combination of hypertension and body size further increases depressive symptoms has not been studied in detail. Since hypertension and obesity correlate positively [15], a similar mechanism may affect depression. Alternatively, hypertension and low body weight affect depression through various mechanisms. It has been reported that undernutrition is associated with depression in underweight [16], whereas lifestyle factors such as stress and poor sleep are related to non-obese hypertension [17]. Therefore, it is hypothesized that hypertension combined with low body weight intensifies depressive symptoms through various mechanisms. Therefore, this cross-sectional study investigated the role of low body weight in the relationship between hypertension and depressive symptoms among middle-aged and elderly individuals living in the community Shika town.” (L83-94)

Comment 8

  1. MATERIAL AND METHOD:

It seems necessary to clarify how the representative sample was calculated and to make a flowchart of the enrollment of respondents and study procedures.

Please add the reasons of inclusion and exclusion respondents

Response 8

We have added Figure 1 (Participant recruitment chart) in the participants section of the revised manuscript (below line 158)

Round 2 Response 8

We have already added it.

Comment 9

  • The instruments description should include the psychometric properties and the reference to the validation studies for the Japanese population. Please add more information about standardized survey instruments.

Response 9

We have amended the assessment of depressive symptoms section of the revised manuscript as follows:

“Each item is rated in a yes/no format. Among them, 10 items (2, 3, 4, 6, 8. 9, 10, 12, 14, and 15) indicate the presence of depression when answered “yes” (positive), while the remaining 5 items (1, 5, 7, 11, and 13) indicated depression when answered “no” (negative). The potential total score ranged from 0 to 15.” (L181-185)

Reliability has been demonstrated in other studies; therefore, the assessment of depressive symptoms section of the revised manuscript was amended as follows:

“The validity and reliability of the GDS-15 Japanese version for depression assessed against the Diagnostic and Statistical Manual of Mental Disorders, Fourth Edition, Text Revision (DSM-â…£-TR) criteria have already been confirmed [23].” (L185-188)

Round 2 Response 9

We further added the following sentence:

“When the cutoff point was set at 6/7, it has been confirmed that the sensitivity was 0.98, specificity was 0.86, and Cronbach's alpha reliability coefficient was 0.83 [25].” (L141-143)

Comment 10

  • Please add sentence about ethical aproval: „The study was conducted according to the guidelines of the Declaration of Helsinki and approved by the Ethics Committee of the …”.

Response 10

We presented the information including the "Institutional Review Board Statement" or the "Informed Consent Statement" at the bottom of the conclusions section from the original file, using the template file format as follows:

“Institutional Review Board Statement: This study was conducted according to the guidelines laid down in the Declaration of Helsinki and all procedures involving research study participants were approved by the Ethics Committee of Kanazawa University (No. 1491).

Informed Consent Statement: Written informed consent was obtained from all participants prior to their participation.” (L480-484)

Round 2 Response 10

We think we have adequately answered the question.

Comment 11

  • Please add the reference to „Body size assessment”

Response 11

We have added the reference no.21.

Round 2 Response 11

We think we have adequately corrected it.

Comment 12

  • Table 1: please explain: SD / % . In this column is SD or %?

Response 12

We have modified Table 1 for clarity.

Round 2 Response 12

We think we have adequately corrected it.

Comment 13

5) RESULTS

  • Each of the results should be mentioned in more depth, indicating references.

Response 13

We have mentioned some interpretations of the results as follows.

“Therefore, the combination of hypertension and low body weight is synergistically related to depressive symptoms only in females, but not in males.” (L306-308)

“Therefore, the multivariate analysis also showed that females with low body weight were associated with depressive symptoms in hypertension, but not in normotension.” (L337-339)

Round 2 Response 13

We think we have adequately corrected it.

Comment 14

  • Please improve all tables. Tables should be easy to interpret and understand.

Response 14

We have modified all Tables for clarity.

Round 2 Response 14

We think we have adequately corrected it.

Comment 15

6) DISSCUSION:

  • The discussion of the results should be done in a more in-depth way.

Response 15

We have added the following sentence to the discussion

“In addition, a cross-sectional study by Hadi et al. reported that central obesity mediates hypertension and depression [26].” (L385-386)

“Although previous studies on body size and depression have implicated both underweight and overweight, one of the differences seems to be due to racial influences. Therefore, we considered it meaningful to conduct epidemiological studies on Japanese people belonging Asian descent.” (L391-395)

Round 2 Response 15

We further amended the Discussion section as follows:

Alternatively, a review by Mendoza et al. [15] discussed the significance of adiposity in obesity hypertension centers on humoral mechanisms through stimulation of the renal-angiotensin system, leptin activity, sympathetic overdrive, and pro-inflammatory processes. Although different factors may be involved in the influence of hypertension on depression, such as physical inactivity, unhealthy lifestyles, including excessive alcohol consumption, and increased sympathetic nerve activity owing to stress, it is necessary to separate these factors in obese and non-obese hypertension. Our findings that showed a significant correlation with GDS in underweight hypertensive females suggest that further investigations are required to determine which factors may synergistically exacerbate depressive symptoms.(L282-291)

“An investigation of healthy Polish women in various stages of menopause by Cybulska et al. [41]. showed that specific genotypes are related to depressive symptoms.” (L336-337)

“Therefore, we speculate that the mechanism why only females with hypertension and low body weight have worsening depressive symptoms may be because of the reduced leptin associated with inadequate nutrition, the increased blood pressure lifestyle with non-obesity, and the reduction in estrogen associated with menopause, synergistically contributing to depression.” (L340-344)

Comment 16

  • I suggest to prepare some kind of summary with gaps indication and future perspective

Response 16

We have added the Limitation and Future Directions section as follows:

“Future studies need to evaluate whether nutritional interventions could improve depressive symptoms in women with hypertension and low body weight.” (L457-458)

Round 2 Response 16

We have further amended Round 1 Response 16 as follows:

“Future research should elucidate the specific combination of factors among low nutrition, lifestyle habits that increase blood pressure, and low estrogen levels that synergistically exacerbate depressive symptoms. Furthermore, it is necessary to evaluate longitudinal studies to determine whether any combination of nutritional guidance, lifestyle guidance for hypertension, and estrogen replacement therapy can effectively alleviate depressive symptoms.” (L352-257)

Comment 17

7) REFERENCES: Please carefully check the references cited in the manuscript. I suggest conducting a new literature review. This article needs the newest references (written after 2020).

Response 17

We have added newest reference: no.13, no.20, no.21, and no.26.

Round 2 Response 17

We have further added the newest references: no.15, no. 17, no. 41.

Comment 18

8) LIMITATION: please add limitation like subsection.

Response 18

We have added the “Limitations and Future Directions” heading at L431.

Round 2 Response 18

We have already corrected it.

Comment 19

  • Please discuss widely the advantages and disadvantages of this review, identify knowledge gaps in the existing literature and try to indicate the needs for future research.

Response 19

We have amended the discussion in the revised manuscript as we replied in responses 15 and 16.

Round 2 Response 19

We have further amended the Limitations and Future Directions as follows:

“Several limitations need to be addressed. Since this was a cross-sectional study, further longitudinal studies are required to elucidate the causal relationship between hypertension, low body weight, and depressive symptoms. Furthermore, nutrient intake and estrogen levels were not evaluated. Another limitation was that depressive symptoms other than GDS were not evaluated. Moreover, depression was not diagnosed. Lastly, many participants are needed in the overweight group for future analyses.

Future research should elucidate the specific combination of factors among low nutrition, lifestyle habits that increase blood pressure, and low estrogen levels that synergistically exacerbate depressive symptoms. Furthermore, it is necessary to evaluate longitudinal studies to determine whether any combination of nutritional guidance, lifestyle guidance for hypertension, and estrogen replacement therapy can effectively alleviate depressive symptoms.” (L346-357)

Comment 20

9) Since some language mistakes mainly typos are in the text, I suggest English editing before publishing of this manuscript

Response 20

The English editing of the present manuscript was done when the original manuscript was written.

Round 2 Response 20

We again had English proofreading of our revised manuscript.
